# Stress Echocardiography Protocol for Deciding Type of Surgery in Ischemic Mitral Regurgitation: Predictors of Mitral Regurgitation Recurrence following CABG Alone

**DOI:** 10.3390/jcm10214816

**Published:** 2021-10-20

**Authors:** Radoslaw Piatkowski, Janusz Kochanowski, Monika Budnik, Michal Peller, Marcin Grabowski, Grzegorz Opolski

**Affiliations:** 1st Chair and Department of Cardiology, Medical University of Warsaw, 02-097 Warsaw, Poland; kochanowski.janusz@gmail.com (J.K.); moni.budnik@gmail.com (M.B.); michalpeller@gmail.com (M.P.); radekp1@wp.pl (M.G.); radekp1@mp.pl (G.O.)

**Keywords:** stress echo, ischemic mitral regurgitation, tenting area, coaptation high

## Abstract

Purpose: Although coronary artery bypass grafting alone (CABGa), or, with mitral annuloplasty (CABGmp), is considered the best therapeutic strategy for patients with ischemic mitral regurgitation (IMR), some recurrences are still reported. The aim of this study was to evaluate the use of the mitral deformation indices (MDI) as a predictor of recurrence of mitral regurgitation in a 12-month follow-up after CABG alone. Methods: A total of 145 patients after myocardial infarction with significant IMR, eligible for CABG, were prospectively enrolled in the study. Mitral valve morphology, left ventricle function, IMR degree as assessed by effective regurgitation orifice area (ERO), myocardial viability, and MDI were assessed prior to surgery. Patients were referred for CABGa (gr.1; *n* = 90) or CABGmp (gr.2; *n* = 55) based on clinical assessment, and the results of rest and stress echocardiography (exercise echocardiography and low dose dobutamine echocardiography-DBX). One year after surgery, each patient underwent the evaluation of cardiovascular events. Univariable logistic regression analysis was used to identify the factors of recurrence of IMR in 1 year follow-up. Serial echo examinations were performed in all patients at discharge, and at 1 and 12 months after surgery. Results: Logistic regression analysis revealed that in CABGa, group preoperative changes of tenting area (TA) and coaptation high (CH) during DBX remained the predictors of the recurrence of IMR in 12 months follow-up. TAdbx > 1 cm^2^ provided a sensitivity of 90% and specificity of 29%, (AUC 0.6436). The best cut-off value for CHdbx was 0.4 cm (sensitivity 90%, specificity 34%; AUC 0.6432). In both groups (CABGa vs. CABGmp) no significant differences were observed in 12-month mortality (1.2% vs. 0%; *p* = 1.0), hospitalizations due to the heart failure (HF) exacerbation (5.9% vs. 8.5%; *p* = 0.72), and in the incidence of the composite endpoint (deaths/CV hosp/stroke) (7% vs. 8.5%; *p* = 0.742). Conclusions: The preoperative assessment of MDI changes during dbx can be used to identify patients with IMR qualified to CABG alone at increased risk of recurrence of IMR in 1 year follow-up. Mitral deformation analysis should be used for a better qualification of patients with IMR to the exact surgical approach.

## 1. Introduction

Ischemic mitral regurgitation (IMR) worsens the prognosis of patients after coronary artery disease, both in the short-term and in the long-term follow-up [1,2,3]. IMR is a dynamic process, and its severity changes with hemodynamic variation. IMR is predominantly related to left ventricle (LV) remodeling and mitral valve deformation [4,5,6,7]. LV spherization, mitral annulus enlargement, increased tenting area (TA) and volume, and the loss of systolic mitral annular contraction all contribute to the development of IMR [8]. The treatment for IMR is still a heavily debated topic with an absence of definitive evidence to support either school of thought. The complex pathomechanism of IMR development is responsible for problems when elaborating efficient therapeutic methods in patients qualified to CABG alone. Recently, due to better understanding of this complex disease and better medical management options for coronary artery disease, the results of surgical treatment have improved. These surgical options include an isolated CABG procedure to ensure revascularization or mitral valve replacement along with CABG as combined procedure or other mitral valve repair techniques with CABG. Restrictive annuloplasty combined with CABG is the most often performed surgical procedure to treat patients with significant IMR. However, the sobering results of the current strategies create the need for a better preoperative assessment of the mitral valve and LV geometry and function [9,10]. A strict correlation between stress-induced (exercise and dobutamine) changes of IMR, mitral deformation indexes (MDI), myocardium viability, clinical symptoms, and prognosis should be considered when evaluating the eligibility of patients with moderate or severe IMR for the appropriate type of surgery [11,12,13,14]. This would help improve risk stratification and the identification of the subgroups of patients with risk of recurrent IMR and worse prognosis who could likely benefit from different surgical strategies. In our opinion, the greater severity of stress induced IMR and changes of MDI correlates with a greater necessity to perform mitral valve repair to avoid the risk of the recurrence of IMR in long-term follow-up.

The aim of this study was to assess the significance of echocardiographic parameters achieved from stress echo (exercise echocardiography and low dose DBX) as a predictor of recurrent IMR in patients with moderate or severe IMR qualified to CABG alone in 12-month follow-up.

## 2. Materials and Methods

### 2.1. Study Population

This prospective observational cohort study included patients aged >18, with a history of myocardial infarction and eligible for CABG [15]. All patients had moderate or severe IMR caused by restrictive systolic leaflet motion (Carpentier’s type IIIb) with or without annular dilatation (Carpentier’s type I). The study was conducted between 2010 and 2017.

A total of 170 potentially eligible patients with IMR qualified to CABG were enrolled in the present study. Of these, 25 patients were excluded from the analysis (7 patients had a poor acoustic window for dobutamine echocardiography; 4 patients in whom viability was not demonstrated within left ventricular segments with impaired contractility; 14 had contraindications for exercise). Finally, 145 patients were qualified to CABGa or CABGmp and prospectively enrolled into the study. (Figure 1). The time frame for enrollment of patients into the study was 24 months.

The patients were divided into groups according to the type of intervention planned: group I (*n* = 90)—CABG alone (CABGa); group II (*n* = 55)—CABG combined with mitral annuloplasty (CABGmp). Additionally, 9 patients were qualified for CABG with mitral valve replacement (CABGmr), but this group was excluded from the analysis. Based on exercise echocardiography (ExE) and dobutamine stress echocardiography (DBX), a diagnostic algorithm for a precise determination of the range of surgical intervention has been elaborated (Table 1). Patients had to meet all criterion to be placed in a given category. Patients were qualified to CABGa if IMR deformation parameters improved during preoperative DBX and were not impaired during EXE (EROexe < 20 mm^2^). The eligibility of patients with an increase in IMR with exercise for CABGa or CABGmp was determined by the results of DBX. The improvement of mitral valve deformation parameters with DBX was the main determinant of eligibility for the CABGa group. On the other hand, CABGmp was considered for patients without an improvement in mitral valve geometry during DBX (Table 1).

The study inclusion criterion was the presence of a significant area of viable myocardium found during DBX (improvement in wall motion of at least four dysfunctional segments). The assessment and presence of myocardial viability was necessary for inclusion in both study groups. The exclusion criteria included a left bundle branch block (LBBB), unstable angina, prosthetic heart valve, other valvular or congenital heart diseases, history of CABG, and severe heart failure (HF) symptoms (NYHA IV—New York Heart Association).

Only patients who underwent complete revascularization were included in the further analysis.

An echocardiographic and clinical assessment was performed at discharge, after 1 month and 12 months.

Each patient signed an informed consent form, and the study was approved by the institutional review board of the Medical University of Warsaw. The study was conducted according to the principles stated in the Declaration of Helsinki.

### 2.2. Surgery

The same surgical team performed surgery through a median sternotomy. Regardless of the proposed way of treatment, the final decision on the surgical technique to be used within the mitral valve was always made by the operating surgeon. However, this did not result in any change in the original treatment group assignment. In all patients, CABG was performed using cardiopulmonary bypass in moderate hypothermia with crystalloid and blood cardioplegia. The main aim of the surgery was to perform a complete coronary revascularization and, in some patients, mitral restrictive annuloplasty. The ring size was determined after measuring of the height of the anterior leaflet and intertrigonal distance, and then downsizing by two sizes (undersizing annuloplasty) [16,17].

The intraoperative criteria for a successful surgery were as follows: leaflet coaptation height (CH) ≤ 0.6 cm, tenting area (TA) ≤ 1.2 cm^2^, and IMR ≤ 1 grade. Recurrent IMR was the insufficiency of at least moderate (ERO > 10 mm^2^) or more at follow-up visits.

### 2.3. Echocardiography

Transthoracic echocardiograms (TTE) were performed within 2–3 days before surgery, and serial TTE examinations were performed at discharge and at follow-up visits. All measurements were performed using the iE33 system (version 4.2–5.0; Philips Medical Systems, N.A., Bothell, WA, USA) equipped with a broadband transducer for a TTE of 2.5–3.5 MHz frequency. IMR severity was assessed by measuring the effective regurgitant orifice area (ERO), with ERO > 10 mm^2^ and <20 mm^2^ considered moderate and ERO ≥ 20 mm^2^ considered severe, as well as mitral regurgitation volume (MRvol) with MR vol ≥ 30 mL considered severe [18,19,20]. ERO and MRvol were calculated using flow convergence (proximal isovelocity surface area-PISA method). The radius of the PISA (r) was measured from the vena contracta level to the point of color Doppler aliasing. ERO was calculated as: 6.28 × r^2^ × Va/Peak V RegJet, where Va is aliasing velocity and VRegJet is the peak velocity of the regurgitant jet by Continuous Wave Doppler. The MRvol was calculated as ERO × VTI RegJet, where VTI RegJet is the VTI of the regurgitant jet [21].

Wall motion abnormalities were evaluated in accordance with the recommendations of the American Society of Cardiology [22]. The wall motion score index (WMSI) was calculated according to a 17-segment model [23]. The left ventricular volumes and ejection fraction (EF) were assessed by the biapical Simpson disk method. The mitral valve deformation was evaluated by measuring the tenting area (TA), i.e., the area enclosed between mitral leaflets and the line of the annular plane and the coaptation height (CH), i.e., the distance between leaflet coaptation and the mitral annular plane from the parasternal long-axis view at mid-systole [24].

### 2.4. Stress Echocardiography

Low-dose DBX was used to distinguish akinetic viable segments from nonviable myocardial regions [25]. The presence of a significant area of viable LV myocardium was the condition for patient inclusion for further analysis. Additionally, during DBX, the dynamics of MDI (increase or decrease) and IMR changes were analyzed. DBX was performed in accordance with current guidelines [26]. For the detection of inotropic response in heart failure patients, we used stages of 5 min, starting from 5 up to 20 mg/kg/min. The next step of qualification for the surgery included ExE to assess the dynamics of IMR changes and TRPG (tricuspid regurgitation pressure gradient as the exponent of the right ventricle overload).

All subjects also underwent a symptom-limited graded exercise echocardiography (ExE) test to assess the dynamics of IMR changes and TRPG, the latter as the exponent of right ventricle overload. The symptom-limited grade ExE was performed according to the following protocol: the initial workload of 25 watts (WAT) was maintained for 3 min, and then the workload was increased every 2 min by 25 W. Blood pressure and a 12-lead electrocardiogram were recorded every 2 min. Two-dimensional and Doppler echocardiographic recordings were available throughout the test. Exercise was interrupted when ischemic electrocardiographic signs, fatigue, or intolerable dyspnea appeared [26].

### 2.5. Clinical End Points

The efficacy of the diagnostic algorithm was evaluated by analyzing the results obtained during the 12-month follow-up CABGa group of patients, which included:functional status (NYHA, CCS—Canadian Cardiovascular Society).dynamics of selected LV parameters changes (EF, WMSI).analysis of the recurrence of moderate or severe IMR.midterm (12 months) mortality; andhospitalization due to exacerbation of HF symptoms.

### 2.6. Data Collection

Baseline clinical characteristics (demographics, medical history, and therapy) and echocardiography (rest, stress) examinations performed at the baseline visit were retrieved from patient’s medical records. Preoperative and postoperative clinical status was determined according to the criteria of the NYHA and the CCS functional class for HF and angina, respectively.

### 2.7. Statistical Analysis

All continuous variables presented non-normal distribution based on a Shapiro-Wilk test and were demonstrated as median values and interquartile ranges. Categorical variables were presented as the number of patients and percentages. Differences between groups for categorical and continuous data were analyzed by Fisher’s exact test and a Mann-Whitney U test, respectively. To establish the predictive values of analyzed variables, univariate logistic regression models were used. Because of the low value of events per variable, multivariate models were not analyzed. Predicting values of variables were presented as ROC curves. All tests were two-tailed, and a *p* value of 0.05 or less was considered statistically significant. All analyses were performed using SAS statistical software, version 9.4 (SAS Institute Inc., Cary, NC, USA).

## 3. Results

### 3.1. Baseline Characteristics

Table 2 and Table 3 show the preoperative clinical and echocardiographic variables in two analyzed groups (CABGa and CABGmp). The patients in the CABGmp group were in a higher NYHA class and had a higher incidence of chronic kidney disease. As expected, patients in the CABGmp group had lower EF, higher WMSI, larger LA, LV dimensions and volumes, and lower EF than patients in the CABGa group. Compared with the CABGa group, significantly higher values of MDI and systolic mitral area were observed in CABGmp. During ExE, significant differences concerning workload were seen in the given groups of patients. Patients in the CABGmp group showed a significant exercise induced IMR and TRPG increase. During DBX, in the CABGa group, the normalization of MDI and a significant decrease in the IMR value were observed (*p* < 0.001); in CABGmp, moderate/severe IMR persisted. During DBX, in the CABGmp group, no significant improvement in MDI was noticed.

### 3.2. In-Hospital Outcomes

We observed significant differences in the rate of in-hospital serious adverse events between the CABGa group and the combined procedure group (Table 4). There was significantly longer length in hospital stay, higher rate of acute kidney disease, respiratory failure, and heart failure events in the combined-procedure group. There was also a higher rate of in-hospital atrial fibrillation (30.3% vs. 51.9%, *p* = 0.0132) in the combined-procedure group than in the CABGa group. In-hospital mortality was higher in the CABGmp group (CABGa vs. CABGmp: 5.6% vs. 16.4%, *p* < 0.0001).

### 3.3. Echocardiographic Results/Recurrence Mitral Regurgitation

After CABG alone, or with mitral annuloplasty, TTE evaluation revealed acceptable results in all the cases: there was no MR or mild MR. Before discharge, TTE was performed, which confirmed the good results of surgical treatment in both groups of patients. Unfortunately, at 1 year, the prevalence of moderate or severe IMR (ERO > 10 cm^2^) was higher in the CABGa group than in the combined-procedure group (25% vs. 17%, *p* < 0.01) (Table 5).

We compared patients with at least moderate IMR after CABGa to patients without or only mild regurgitation one year after surgery (Table 6). To explore the effect of revascularization on the risk of recurrence of IMR in CABGa group, we analyzed changes in WMSI, which were stratified according to the recurrence of IMR defined as an ERO >10 mm^2^ in 1-year follow-up. The improvement in the global WMSI was significantly higher for patients who were free of moderate or severe IMR at 1 year than for those with such mitral regurgitation (1.3 vs. 1.6, respectively, *p* = 0.0109). In the CABGa group, patients who never had recurrence of moderate or severe IMR and who had not undergone a mitral-valve intervention had lower LV volumes and EF than those with recurrence of IMR (ESV, 56.1 and 83.5 mL, respectively; *p* = 0.002; EF, 51.2 and 42.6%, respectively, *p* = 0.0005). The comparison of clinical and echocardiographic data in patients with IMR with ERO ≤ 10 mm^2^ and ERO > 10 mm^2^ in 1-year follow-up is presented in Table 6.

Univariate analysis results with respect to the recurrence of moderate or severe IMR in the CABGa group are shown in Table 7. Left atrial dimensions and MDI’s changes (TA and CH) during DBX before surgery were significant predictors of the presence of at least moderate IMR in the CABGa group in a 12-month follow-up. Preoperative TAdbx and CHdbx were the independent predictors of the risk of recurrence of IMR. TAdbx > 1 cm^2^ provided a sensitivity of 90% and specificity of 29% (AUC 0.6436). The best cut-off value for CHdbx was 0.4 cm (sensitivity 90%, specificity 34%; AUC 0.6432) (Figure 2 and Figure 3). None of the exercise echo parameters predicted the recurrence of IMR in a 12-month follow-up.

### 3.4. Clinical Outcomes

At 1 year, we observed no significant difference in death rates between the study groups, with 1.2% for CABGa and 0% for the combined procedure (*p* = 1.000). Overall rates of cardiovascular rehospitalizations did not differ significantly in the two study groups (CABGa vs. CABGmp: 5.9% vs. 8.5%, *p* = 0.72). In addition, there were no significant differences in the composite endpoint of major adverse cardiac or cerebrovascular events (death/hospitalization/stroke) between the study groups (CABGa vs. CABGmp: 7% vs. 8.5%; *p* = 0.742) (Table 5).

## 4. Discussion

In our study, the qualification of patients for CABG alone was based on very strict echocardiographic criteria, the crucial element of which was to reveal the myocardial viability of those LV segments, the dysfunction of which during rest examination generated IMR. A cumulative analysis of ExE and DBX results allowed for the final therapeutic decisions. Simultaneously, an indispensable condition was to obtain a complete normalization of MDI and a decrease in IMR grade during DBX. Roshanali et al. used only simple qualitative echocardiographic criteria based on the 4-grade scale of mitral regurgitation to identify patients with moderate IMR. They showed the utility of low-dose DBX in selecting patients who would be undergoing CABG to receive concurrent mitral valve repair {13}. We have developed a more advanced protocol based on exercise echocardiography (ExE) and dobutamine stress echocardiography (DBX) criteria for the precise determination of the range of surgical interventions. The presented criteria allowed us to qualify 90 patients for this type of treatment. Unfortunately, postoperative results showed that some patients in the CABGa group have at least moderate IMR in 1 year after surgery. The main aim of the study was to predict failure using preoperative information, and providing the clinician with guidance regarding the choice of therapy. In the CABGa group, a statistically significant IMR reduction was found in most of patients in follow-up. The complete revascularization performance resulted in significant improvement in LV geometry and function. A reduction in the degree of mitral regurgitation with CABG alone has been reported previously [27,28,29]. Penicka et al. found that in a series of patients with moderate IMR who underwent CABG alone, the resolution of MR after surgery was associated with more viable segments and less LV desynchrony at baseline [30]. Kang et al. reported that patients who demonstrated an improvement in LV function and a reduction in LV size after CABG also had a reduction in the IMR grade one year after surgery [28]. In our study, we observed a similar reduction in the IMR in the CABGa group. The progress of LV remodeling and, secondarily, the posterior mitral valve leaflet restriction, is a mechanism responsible for the lack of improvement or increase in the IMR grade after surgery [31]. In our study, we observed a decrease in LV volumes and diameters, as well as an increase in EF in most patients; favorable echocardiography results were reflected in the patient’s clinical status. Moreover, the absence of IMR after surgery was associated with improvement in global wall-motion scores at 1 year. Michler et al. randomly assigned 301 patients with moderate IMR to undergo either CABG alone or the combined procedure with mitral repair. They reported that improvements in both global and regional wall motion scores and the presence of LV reverse remodeling were associated with significantly less moderate or severe mitral regurgitation at a 2 year follow-up [32]. Improvement in global wall motion score indexes, decreases in LV volumes, and increases in EF after revascularization are indicative of viable myocardium. As a result, it can also improve mitral valve function in patients with IMR in relation to the decrease in LV size, increased mitral valve closing forces, improved papillary-muscle synchrony, and enhanced myocardial contractility. Therefore, surgical decision making could be improved by identifying which patients are most likely to have an improvement in LV function after revascularization, which can lead to a postoperative reduction in IMR. In our study, in all patients, we assessed both myocardial viability and changes in mitral valve geometry (tenting area and coaptation heigh) using low-dose DBX and exercise echocardiography. We observed significant correlations between preoperative changes in MDI and the presence of significant IMR 1 year after surgery by using DBX, but not exercise echocardiography. These findings suggest that DBX-induced reversible ischemia changes, especially in the posterior wall, could improve MDI. Improvement in wall motion contractility in segments supporting the posterior papillary muscle may reduce the degree of IMR by decreasing leaflet tethering forces that cause incomplete mitral leaflet closure. Abe et al. reported similar observations in a small group of patients with coronary artery disease and moderate IMR [33]. Our observations also suggests that the presence of contractile reserve, as well as improvement in MDIs, identified with DBX, may serve as a predictor of reduced IMR in response to revascularization in patients qualified to CABG.

Despite the higher proportion of patients with moderate or severe IMR at 1 year in the CABGa group, 1-year clinical outcomes, including functional status, mortality, and major adverse cardiac and cerebrovascular events, did not differ significantly between the study groups. Our results are like those of one randomized trial involving patients with moderate ischemic mitral regurgitation [34]. However, our study was not powered to detect small, but important differences in survival and clinical composite endpoints.

The results of our study indicate that some patients experience a recurrence of IMR after surgery. Currently, in patients with a recurrence of significant IMR after CABGaor CABGmp (especially after failed surgical mitral valve repair), MitraClip may be a good option [35]. However, our study was conducted between 2010 and 2017, when MitraClip interventions were not as available as they are today.

To our knowledge, this is the first prospective case-series analysis of the patient’s qualification for surgical treatment based on all essential elements of mitral complex functioning in echocardiography examination at rest, as well as during stress echo.

The main limitation of the study was a relatively low number of included patients. Therefore, the statistical power of correlations is lowered. Because of the low value of events per variable, multivariate models were not analyzed. It must be stressed, however, that significant IMR in patients treated with CABGa is seen only in a small number of patients. Taking into consideration the innovative character of this study, the collected group of patients belongs to the most numerous presented in the literature, and is the first with such a complex methodology of a patient’s qualification.

Another limitation of the study is the lack of randomization, which may result in potential selection bias, possibly leading to incorrect conclusions. It must be stressed that in the present study, deterministic criteria for patients’ qualification to the suitable type of treatment were used. A lack of randomization of patients resulted in uneven selection to the CABGa and CABGmp groups.

Our study results indicate that the application of an elaborated diagnostic algorithm may improve the qualification of patients with significant IMR for a suitable type of surgical procedure, finally enabling good clinical results in the 12-month follow-up after surgery. Based on such qualification in selected patients, the CABG alone can ameliorate IMR and produce beneficial functional and structural improvements without an increase in long-term mortality. The current guidelines emphasize that in patients with severe IMR, valve surgery is recommended in patients undergoing CABG. In selected patients without advanced LV remodeling, mitral valve repair is recommended. In contrast, additional valve replacement may be considered in patients with echocardiographic predictors of repair failure.

The guidelines also emphasize the importance of exercise echocardiography, which may help to identify patients with severe mitral regurgitation when echocardiography at rest is inconclusive [36]. Our study included the use of exercise echocardiography in the qualification protocol to evaluate changes in IMR during an exercise. The guidelines highlight the difficulties in managing patients with moderate IMR. They emphasize that surgery is more likely to be considered if myocardial viability is present, and if comorbidity is low. In addition, an exercise-induced large increase in mitral regurgitation severity and systolic pulmonary artery pressure favors combined surgery (CABG with mitral intervention) [35]. Our study protocol included an assessment of all these elements (myocardial viability and exercise induced IMR changes). In addition, it included the assessment of changes in the geometry of the mitral valve during DBX, which is important in the pathomechanism of IMR.

Because this novel clinical decision-making tool has been applied in a small group of patients, the present results must be considered hypothesis generating.

This must be clearly stated, along with the need to study a larger group before broadly implementing the approach. Certainly, further validation and randomized studies with more patients enrolled and followed-up for a longer period are necessary.

In conclusion, it should be emphasized that CABG alone could be contemplated if IMR parameters improve with preoperative DBX, and do not impair with EXE. Moreover, CABGmp could be considered for patients without IMR improvement during DBX, and with deterioration during ExE. That decision should be individualized for patients with just one stress test (DBX or EXE) showing positive findings.

This study has highlighted the importance of detailed preoperative TTE examination (rest and stress echo) in patients undergoing surgery for IMR. It can be used to identify patients with IMR who are likely to have recurrent IMR after CABG. Specifically, higher LV volumes, lower EF, higher tethering area at rest TTE, and changes of MDI indexes during preoperative DBX were associated with a recurrence of IMR. In these patients, changes and additional repair techniques or MV replacement should be considered.

## Figures and Tables

**Figure 1 jcm-10-04816-f001:**
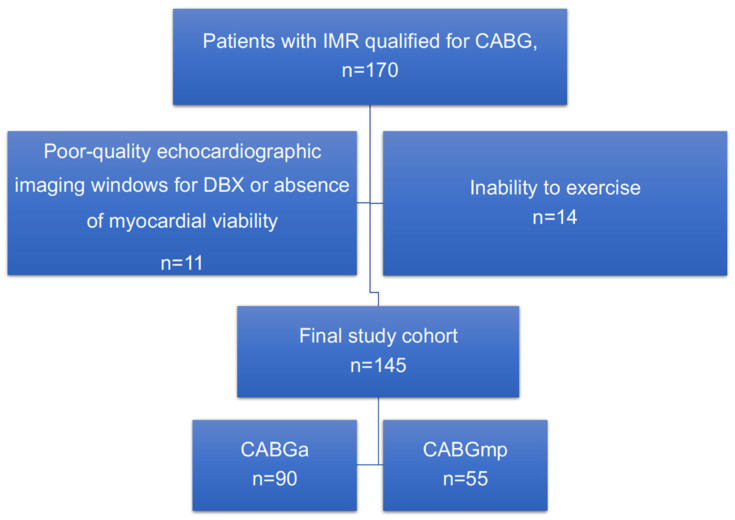
Study flowchart. CABG—coronary artery bypass grafting; DBX—dobutamine stress echocardiography.

**Figure 2 jcm-10-04816-f002:**
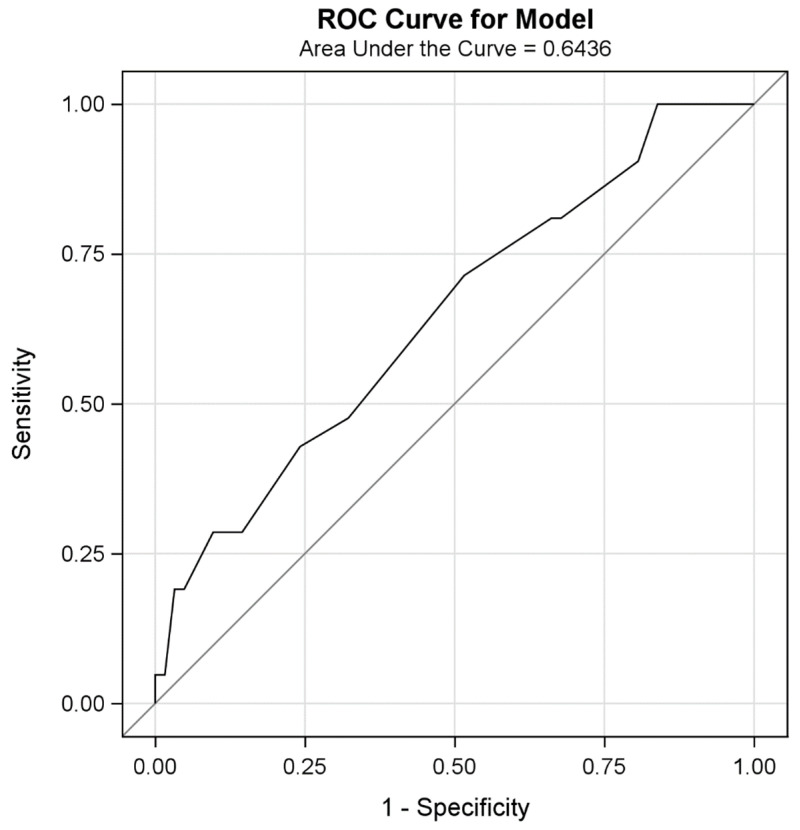
Receiver operator characteristic (ROC) curve for TA during DBX as a predictor of mitral regurgitation with ERO > 10 mm^2^ in patients 12 month after CABG.

**Figure 3 jcm-10-04816-f003:**
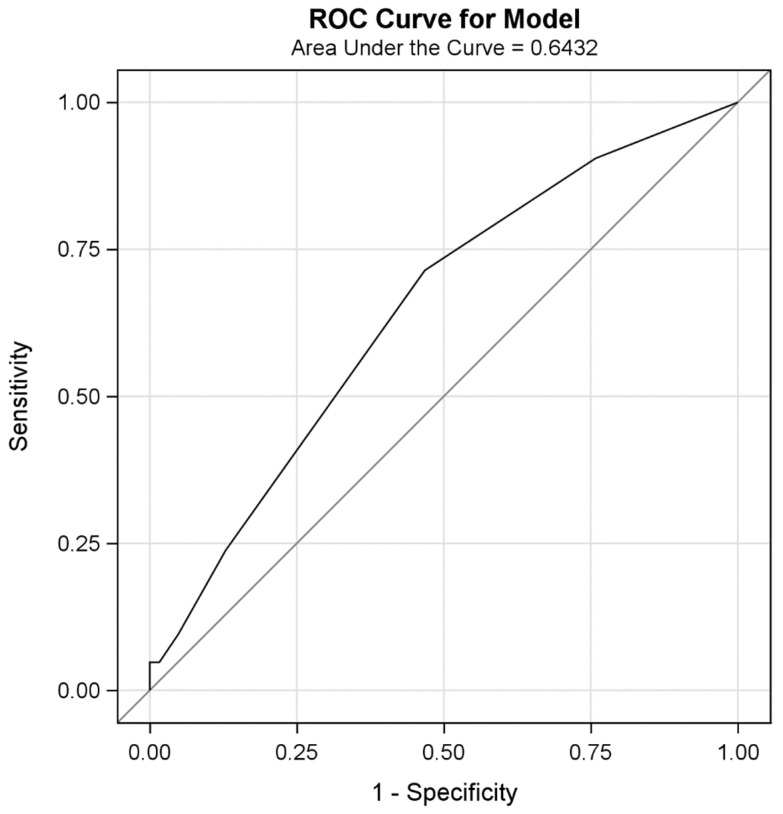
Receiver operator characteristic (ROC) curve for CH during DBX as a predictor of mitral regurgitation with ERO > 10 mm^2^ in patients 12 month after CABG.

**Table 1 jcm-10-04816-t001:** Eligibility criteria to a given type of surgery.

	CABGa	CABGmp	CABGmr (Not Included in Analysis)
ERO exe	<20 mm^2^ or ≥20 mm^2^	≥20 mm^2^	≥20 mm^2^
CH dbx	≤6 mm	6 mm < CH ≤ 10 mm	CH > 10 mm
TA dbx	≤1.2 cm^2^	1.2 cm^2^ < TA ≤ 2.5 cm^2^	TA > 2.5 cm^2^
ERO dbx	<10 mm^2^	- *	- *

exe—exercise echocardiography; dbx—dobutamine echocardiography; CABGa—coronary artery bypass grafting alone; CABGmp—coronary artery bypass grafting with mitral annuloplasty; CABGmr—coronary artery bypass grafting with mitral replacement; WMSI—wall motion score index; ERO—effective regurgitant orifice area; TA—tenting area; CH—coaptation height; *—insignificant in qualification strategy.

**Table 2 jcm-10-04816-t002:** Baseline characteristic of patients with significant MR treated by CABGa or CABGmp.

	CABGa *n* = 90	CABGmp *n* = 55	*p* Value
Male sex, *n* (%)	50 (55.6%)	31 (56.4%)	1.0000
BMI (kg/m^2^)	26.9 (17.6–37.5)	26.4 (17.8–35.6)	0.4302
Hypertension, *n* (%)	58 (64.4%)	39 (70.9%)	0.4703
Diabetes, *n* (%)	27 (30%)	22 (40%)	0.2777
Hyperlipidemia, *n* (%)	58 (64.4%)	26 (47.3%)	0.0563
Smoking, *n* (%)	61 (68%)	37 (67.3%)	0.8558
Family history of CAD, *n* (%)	44 (48.9%)	24 (43.6%)	0.6081
Chronic kidney disease, *n* (%)	10 (11.1%)	14 (25.5%)	0.0368
Atrial fibrillation, *n* (%)	15 (16.7%)	16 (29.1%)	0.0956
COPD, *n* (%)	8 (8.9%)	7 (12.7%)	0.5758
NYHA	1.9 (0–4)	2.4 (0–4)	0.0032
One vessel CAD, *n* (%)	2 (2.2%)	10 (10.9%)	0.0535
Two vessels CAD, *n* (%)	22 (24.4%)	14 (25.5%)	1.0000
Three vessels CAD, *n* (%)	66 (73%)	35 (63.6%)	0.2664
Affected vessel treated by CABG, *n* (%) LMCA LAD Cx RCA	21 (23.3%) 89 (98.9%) 70 (77.8%) 78 (86.7%)	14 (25.5%) 47 (85.5%) 39 (70.9%) 47 (85.5%)	0.8423 0.0019 0.4288 1.0000

Data are presented as median values and interquartile ranges, or number (percentage) as shown; CAD—coronary artery disease; BMI—body mass index; COPD—chronic obstructive pulmonary disease; NYHA—New York Heart Association; LMCA—left main coronary artery; LAD—left anterior descending artery; Cx—circumflex artery; RCA—right coronary artery.

**Table 3 jcm-10-04816-t003:** Baseline echocardiographic parameters in patients with IMR treated by CABGa or CABGmp.

	CABGa *n* = 90	CABGmp *n* = 55	*p* Value
Rest echocardiography			
EF (%)	43.6 (18–65)	39.7 (20–60)	0.0184
WMSI	1.6 (1.1–2.7)	1.7 (1.2–2.5)	0.0024
LVDD (mm)	53.6 (43 –69)	56.4 (44–72)	0.0130
LVDS (mm)	40.1 (26–58)	43.8 (30–64)	0.0079
EDV (mL)	125.6 (47–283)	144 (66–252)	0.0265
ESV (mL)	75.1 (18–213)	89.8 (27–174)	0.0363
LA (mm)	41.6 (30–58)	44.3 (34–54)	0.0022
RV (LAX) (mm)	25 (19–36)	26.8 (20–38)	0.0018
SIs	0.38 (0.17–0.58)	0.41 (0.16–0.9)	0.1120
SId	0.49 (0.23–0.77)	0.52 (0.25–0.94)	0.1993
VC (mm)	5.2 (4–8)	6.8 (1–13)	<0.0001
PISA (mm)	6.2 (4–8)	7.5 (5–11)	<0.0001
ERO (mm^2^)	15.6 (11–30)	23.3 (11–49)	<0.0001
MR vol (mL)	23.9 (12–43)	35.9 (17–72)	<0.0001
Coaptation height (mm)	7 (4–14)	10 (6–14)	<0.0001
Tenting area (cm^2^)	1.9 (1.2–3.8)	2.7 (1.5–5.7)	<0.0001
SMA (cm^2^)	9.8 (7.1–13.1)	10.7 (7.2–14.8)	0.0008
Dobutamine echocardiography, *n* (%)	90 (100%)	55 (100%)	
EF (%)	51.3 (25–77)	45.4 (24–64)	0.0010
WMSI	1.3 (1–2.1)	1.5 (1–2.3)	0.0004
TRPG (mmHg)	23.4 (0–44)	31.5 (12–54)	<0.0001
PISA (mm)	3.9 (2–7)	6.8 (2–10)	<0.0001
ERO (mm^2^)	8 (3–16)	19.8 (0.06–43)	<0.0001
MR vol (mL)	13.2 (4–33)	29.7 (2–60)	<0.0001
Tenting area (cm^2^)	1.2 (0.6–3.3)	2.4 (1.3–4.8)	<0.0001
SMA	8.6 (5.8–12)	9.6 (6.7–13.9)	0.0003
Exercise echocardiography, *n* (%)	90 (100%)	55 (100%)	
Workload (Watts)	63.5 (25–100)	60.9 (25–100)	*p* = 0.434
EF (%)	45.5 (19–65)	38.3 (18–64)	<0.0001
TRPG (mmHg)	30.0 (8–69)	48.2 (25–81)	<0.0001
PISA (mm)	5.9 (2–9)	8.9 (7–12)	<0.0001
ERO (mm^2^)	15 (4–33)	32.7 (19–79)	<0.0001
MR vol (mL)	22.9 (5–56)	46.8 (25–119)	<0.0001

Data are presented as median values and interquartile ranges. TAPSE—tricuspid annular plane systolic excursion; rest—echo examination at rest; exe—exercise; dbx—dobutamine echocardiography; MR—mitral regurgitation; WMSI—wall motion score index; VC—vena contracta; PISA—proximal isovelocity surface area; ERO—effective regurgitant orifice; MRvol—mitral regurgitation volume; TA—tenting area; CH—coaptation height; SMA—systolic mitral area; EF—ejection fraction; LVDD—left ventricular end-diastolic dimension; LVDS—left ventricular end-systolic dimension; SIs—sphericity index at end-systole; SIs—sphericity index at end-diastole; EDV—left ventricular end-diastolic volumes; ESV—left ventricular end-systolic volume; TRPG—maximal tricuspid regurgitant peak gradient.

**Table 4 jcm-10-04816-t004:** In-hospital complications in patients with IMR treated by CABGa or CABGmp.

	CABGa *n* = 90	CABGmp *n* = 55	*p* Value
Length of hospitalization (days)	20.5 (4–80)	29.9 (2–114)	0.0069
Atrial fibrillation, *n* (%)	27 (30.3%)	28 (51.9%)	0.0132
Ventricular tachycardia/Ventricular fibrillation, *n* (%)	2 (2.3%)	0 (0%)	0.5266
TIA/stroke, *n* (%)	0 (0%)	2 (3.7%)	0.1409
Infection, *n* (%)	22 (24.7%)	19 (35.2%)	0.1881
Acute kidney disease on dialysis, *n* (%)	4 (4.5%)	14 (25.9%)	0.0004
Respiratory failure, *n* (%)	1 (1.1%)	11 (20.4%)	<0.0001
IABP, *n* (%)	11 (12.4%)	29 (52.7%)	<0.0001
Cardiogenic shock, *n* (%)	4 (4.5%)	11 (20.4%)	0.0041
Need of antiarrhythmic medication use, *n* (%)	23 (25.8%)	24 (44.4%)	0.0277
Bleeding, *n* (%)	1 (1.1%)	4 (7.4%)	0.0674
Death, *n* (%)	5 (5.6%)	9 (16.4%)	0.0443

Data are presented as median values and interquartile ranges, or number (percentage) as shown; TIA—transient ischemic attack; IABP—intra-aortic balloon pump.

**Table 5 jcm-10-04816-t005:** Follow-up after 1-year post surgery.

	CABGa *n* = 90	CABGmp *n* = 55	*p* Value
Recurrence of IMR (ERO > 10 cm) *n*, (% patients)	28 (25%)	9 (17%)	*p* < 0.01
Hospitalization for cardiovascular reasons, *n* (%)	5 (5.9%)	5 (8.5%)	0.72
Death, *n* (%)	1 (1.2%)	0 (0%)	1.0000
Death/hospitalization/stroke, *n* (%)	6 (7%)	5 (8.5%)	0.742
PISA–difference from baseline (mm)	−2.3 (−6.0–+2.0)	−4.9 (−9.0–+1.0)	<0.0001
ERO–difference from baseline (mm^2^)	−6.7 (−25–+12)	−16.8 (−45–+5)	<0.0001
MRvol–difference from baseline (mL)	−9.4 (−37–+28)	−25.9 (−57–−3)	<0.0001

Data are presented as median values and interquartile ranges, or number (percentage) as shown. PISA—proximal isovelocity surface area; ERO—effective regurgitant orifice; MRvol—mitral regurgitation volume.

**Table 6 jcm-10-04816-t006:** Comparison of patients in CABGa group with IMR with ERO ≤ 10 mm^2^ and ERO > 10 mm^2^ in 1 year follow-up.

	ERO ≤ 10 mm^2^ *n* = 62 (75%)	ERO > 10 mm^2^ *n* = 28 (25%)	*p* Value
Male sex, *n* (%)	35 (56.5%)	16 (57.1%)	1.0000
BMI (kg/m^2^)	27.1 (17.6–37.5)	25.9 (18–31.6)	0.2356
Current smoking, *n* (%)	6 (9.7%)	9 (33.3%)	0.0163
Hypertension, *n* (%)	41 (66.1%)	17 (61.9%)	0.7936
Diabetes, *n* (%)	43 (69.4%)	8 (28.6%)	1.0000
Hyperlipidemia, *n* (%)	43 (69.4%)	13 (47.6%)	0.1136
Atrial fibrillation, *n* (%)	10 (16.1%)	5 (19.1%)	0.7444
COPD, *n* (%)	4 (6.5%)	4 (14.3%)	0.3618
NYHA class	1.2 (0–3)	1.5 (1–3)	0.0336
CCS class	1.0 (1–2)	1.1 (1–2)	0.0201
Medications, *n* (%): Beta-adrenolytics ACE-I CC blockers Loop diuretics Statins ASA	59 (95.2%) 51 (82.3%) 12 (19.4%) 50 (80.7%) 59 (95.2%) 61 ((98.4%)	27 (95.2%) 24 (85.7%) 1 (4.8%) 24 (85.7%) 25 (90.5%) 8 (90.5%)	1.0000 1.0000 0.1683 0.7500 0.5965 0.1562
LVDD (mm)	50.5 (41–62)	54.5 (39–68)	0.0342
LVDS (mm)	35.9 (22–51)	41.2 (26–58)	0.0062
EDV (mL)	105.8 (61–220)	132.5 (48–244)	0.0634
ESV (mL)	56.1 (21–150)	83.5 (20–188)	0.0223
LA (mm)	40.3 (30–54)	45.7 (35–65)	0.0042
EF (%)	52.2 (28–70)	42.6 (25–60)	0.0005
WMSI	1.3 (1–1.94)	1.6 (1.1–2.5)	0.0109
Workload (Watts)	92.7 (25–125)	76.1 (25–125)	0.0109

Data are presented as median values and interquartile ranges or number (percentage) as shown; CAD, coronary artery disease; BMI, body mass index; COPD, chronic obstructive pulmonary disease; CCS, Canadian Cardiovascular Society; NYHA, New York Heart Association; ACE-I, angiotensin converting enzyme inhibitors; CC blockers, calcium channel blockers; ASA, acetylsalicylic acid; WMSI, wall motion score index; EF, ejection fraction; LVDD, left ventricular end-diastolic dimension; LVDS, left ventricular end-systolic dimension; LA, left atrium; EDV, left ventricular end-diastolic volumes; ESV, left ventricular end-systolic volume.

**Table 7 jcm-10-04816-t007:** Logistic regression analysis of predictors of ERO > 10 mm2 in CABGa group.

Variable	Univariate Analysis		
	OR	95% CI	*p* Value
Female sex	1.029	0.379–2.794	0.9560
BMI	920	0.802–1.055	0.2349
Smoking	4.667	1.354–16.090	0.0147
Hypertension	0.832	0.298–2.322	0.7258
Diabetes	0.840	0.283–2.489	0.7531
Hyperlipidemia	0.402	0.146–1.105	0.0774
Chronic kidney disease	0.711	0.139–3.646	0.6822
Atrial fibrillation	1.224	0.339–4.411	0.7578
COPD	2.417	0.494–11.822	0.2760
One vessel disease	3.050	0.182–51.039	0.4379
Two vessels disease	0.441	0.115–1.691	0.2326
Three vessels disease	0.889	0.294–2.685	0.8346
Beta-adrenolytic at discharge	1.017	0.100–10.340	0.9887
ACEI at discharge	1.294	0.324–5.170	0.7153
CC blockers at discharge	0.208	0.025–1.710	0.1441
Loop diuretics at discharge	1.440	0.364–5.695	0.6034
Statins at discharge	0.483	0.075–3.110	0.4438
ASA at discharge	0.156	0.013–1.814	0.1377
Rest TTE before surgery			
EF	0.960	0.960–1.014	0.1468
WMSI	2.330	0.504–10.774	0.2790
LVDD	1.039	0.961–1.123	0.3351
LVDS	1.027	0.963–1.095	0.4200
EDV	1.006	0.996–1.015	0.2315
ESV	1.007	0.996–1.018	0.2310
LA	1.107	1.001–1.225	0.0475
TRPG	1.057	0.992–1.126	0.0855
VC	0.700	0.337–1.451	0.3372
PISA	1.297	0.804–2.093	0.2871
ERO	1.33	0.008–18.45	0.1876
MR volume	1.077	0.998–1.163	0.0573
SIs	30.790	0.172–68.23	0.1952
SId	4.253	0.051–351.768	0.547
Coaptation height	6.262	0.436–89.843	0.1770
Tenting area	2.955	0.958–9.113	0.0593
Dobutamine TTE before surgery			
EF	0.961	0.913–1.012	0.1337
WMSI	4.384	0.727–26.455	0.1070
PISA	1.561	1.010–2.414	0.0450
ERO	1.23	0.039–23.1	0.1115
MR volume	1.103	0.993–1.225	0.0669
Coaptation height (CHdbx)	5.52	1.037–92.7	0.0480
Tenting area (TAdbx)	6.307	1.190–33.424	0.0304
Exercise echocardiography before surgery			
EF	0.958	0.908–1.011	0.1184
TRPG	1.040	0.996–1.086	0.0757
PISA	1.243	0.938–1.647	0.1306
MR volume	1.044	0.985–1.106	0.1447

CAD, coronary artery disease; BMI, body mass index; COPD, chronic obstructive pulmonary disease; CCS, Canadian Cardiovascular Society; NYHA, New York Heart Association; ACE-I, angiotensin converting enzyme inhibitors; CC blockers, calcium channel blockers; ASA, acetylsalicylic acid; WMSI, wall motion score index; EF, ejection fraction; LVDD, left ventricular end-diastolic dimension; LVDS, left ventricular end-systolic dimension; LA, left atrium; EDV, left ventricular end-diastolic volumes; ESV, left ventricular end-systolic volume; rest, echo examination at rest; exe, exercise; dbx, dobutamine echocardiography; VC, vena contracta; PISA, proximal isovelocity surface area; ERO, effective regurgitant orifice; MRvol, mitral regurgitation volume; TA, tenting area; CH, coaptation height; Sis, sphericity index at end-systole; Sis, sphericity index at end-diastole TRPG, maximal tricuspid regurgitant peak gradient; LA, left atrium; MR, mitral regurgitation.

## Data Availability

Additional data are available upon request from the first author (R.P.).

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
