# Peer review of "Stress Echocardiography Protocol for Deciding Type of Surgery in Ischemic Mitral Regurgitation: Predictors of Mitral Regurgitation Recurrence following CABG Alone"

_jcm, 2021, doi:10.3390/jcm10214816_

Round 1

Reviewer 1 Report

see the archive below, i cannot paste here...

Author Response

Author’s reply:

The authors describe their protocol for deciding the type of surgery in patients with significant MR and LV dysfunction. I found the protocol, results, and some of the findings interesting, although as the authors agree, hypothesis generating. This is the strength of the study; the weakness is the quite disorganized manner in with the results are presented.

  1. Thus, I think that the main aim of the study should not be“ … to evaluate use of the mitral deformation indices (MDI) as a predictor of recurrence of mitral regurgitation in 12- 16month follow-up after surgery” because this in only demonstrated for the CABG alone, but rather to describe the use and results of their protocol for deciding what to do with these patients…..Now, if the authors want to be stuck to the MR deterioration predictors issue in patients without Mitral interventions, then the subgroup with Mitral intervention has nothing to do with this research and should be excluded.

Answer:

In our study, we presented both groups of patients (CABGa and CABGmp) to show their baseline characteristics and the results of cardiac surgery (clinical and echocardiographic) with particular emphasis on recurrent mitral regurgitation. The number of patients with recurrent IMR was small in both groups. Therefore, we analyzed only the CABGa group - initially more numerous. The CABGmp group with recurrent IMR was too small to analyze. It is necessary to study a large group of patients to obtain adequate statistical power in the CABGmp group.

We slightly changed the title to highlight the CABG group: "Usefulness of stress echocardiography in patients with ischemic mitral regurgitation and left ventricular dysfunction qualified to coronary artery bypass grafting.”

2- The main issue I would like to have clarified is: from patients with significant MR and LV disfunction, how many were sent to CABGa or CABG plus mitral annuloplasty, based on Dobutamine and ExE results. I think that the authors should built a figure showing this. In the figure should also be depicted how many initially considered patients were excluded due to absence of viability of incapacity to exercise on a bicycle.

Answer: We prepared a figure showing this (Figure 1. Study flowchart).

Also, show what is going on with a patient that has mitral tenting and MR improvement with both Dobutamine and exercise.

Answer: Table 1 presents such data.

Or the opposite situation?. How many patients were in these situation?. The total number of patients with the 4 possible scenarios with the decisions taken should be depicted (Dob+ plus ExE+; Dob- plus ExE -; and the 2 intermediate situations).

3- Regarding this point, in the Discussion section should be stated that CABG alone could be contemplated if MR parameters improve with dobutamine and do not impair with exercise, and that CABG-mitral surgery could be considered for patients without MR dobutamine improvemet and with ExE deterioration; but that decision should be individualized for patients with just 1 tecnique showing positive findings.

Answer: Thank you for this valuable attention. We have included this excerpt in the summary in the discussion section.

4- One of the strength of the study in my opinion is that performing this protocol, results were similar in terms of mortality in these 2 non-randomized subgroups, even though the clinical characteristics were worse for the CABG+mitral surgery group. I thing that this point should be enlightened. Also, mitraclip intervention could be reserved for patients with CABGa in case of deterioration of an initially considered non severe MR.

Answer:

Our study was not powered to detect small but important differences in survival and clinical composite endpoints. So, we didn’t enlighten these results. As we highlighted in the discussion, the present results must be considered hypothesis generating.

We agree that there is currently an additional therapeutic option (mitralclip) for patients with recurrent significant mitral regurgitation in follow-up.

5- Not sure if table 6 and discussion regarding MRO more and less than 10 in the CABGa group adds to this paper

Answer: We consider the data in Table 6 to be important because they show differences between the two groups of patients (no recurrence and recurrent mitral regurgitation) and indicate parameters that may play a significant role in the occurrence of recurrence of significant mitral regurgitation (low EF, larger dimensions and left ventricular volume).

6- Consider the inclusion of these references I miss:

- Peteiro et al. Prognostic value of mitral regurgitation assessment during exercise echocardiography in patients with left ventricular dysfunction: a follow-up study of 1.7 +/- 1.5 years. Eur J Echocardiogr. 2008 Jan;9(1):18-25.

- Peteiro et al. Value of resting and exercise mitral regurgitation during exercise echocardiography to predict outcome in patients with left ventricular dysfunction. Rev Esp Cardiol. 2007 Mar;60(3):234-43

-Peteiro et al. Prognostic value of mitral regurgitation assessment during exercise echocardiography in patients with known or suspected coronary artery disease. J Am Soc Echocardiogr. 2006 Oct;19(10):1229-37.

We added:

 Peteiro et al. Prognostic value of mitral regurgitation assessment during exercise echocardiography in patients with left ventricular dysfunction: a follow-up study of 1.7 +/- 1.5 years. Eur J Echocardiogr. 2008 Jan;9(1):18-25.

  1. Tables: Depict parameters at rest and at exercise for Dobutamine and ExE studies (WMSI, MR parameters.etc). Also state functional capacity in Watts.

Answer: All available data are presented in table 3

Reviewer 2 Report

This study’s aim is to assess significance of echocardiographic parameters achieved from stress echo and low dose dbx as a predictor of recurrent IMR in patients with moderate or severe IMR qualified to CABG alone in 12-month follow-up. Patient selection criteria for combined CABG and MVP is a long-standing and important issue, and the use of stress echocardiography is considered as a useful method for selection. Some comments and question.

# Abstract

  • Please use full name of TA, CH in the abstract.
  • The mitral deformation indices (MDI) generally include several parameters. Please specify parameters which are assessed in this study.

# Introduction

  • Line 59, “approptiate surgical [11-13]”. Require correction.

# M&M

  • This study included not only moderate but also severe degree ischemic mitral regurgitation (IMR). This is a unique point. In the reference 19 and other studies, severe IMR generally goes to combination of MVP or other tract, and other studies included moderate IMR in the study population. I wonder the reason the authors include both moderate and severe IMR in the same population.
  • The Eligible criteria suggested by the authors have no reference. Please provide backgrounds of the eligible criteria. Other studies (Roshanali et al. Echocardiograhy 2006;23:31-37 and J Thorac Cardiovasc Surg 2014;148:1323-1327) used similar but more simple eligibility criteria.
  • Line 110, please use the same scale of ERO, (in some cases the authors used 10 mm2 and in some cases 0.1 cm2)
  • In the section of 2.3, “IMR severity was assessed by measuring the effective regurgitant orifice area (EROA), with EROA >10 mm² and <20 mm² considered moderate and ERO ≥20 mm² considered severe as well as mitral regurgitation volume (MRvol) with MR vol ≥ 30ml considered severe [17-19].” What is the background of ERO > 10 mm2 of moderate IMR?
  • Line 117, EROA and ERO looks the same term.
  • Line 129, “The mitral valve deformation” is the same term of “MDI?”
  • Line 138, “DBX”, “dbx”, and “DSE” looks the same term.
  • In the section of 2.4, Please describe the dobutamine doses for the DBX.

# Results

  • In the table2, CABGa group has higher LCx lesions. IMR is known to be related to posterolateral wall involvement. Is the more LCx lesion related to poor outcome?
  • The author used “recurred” IMR. Was the IMR recurred or persisted after surgery? Other studies like Roshanali or Yin (Heart Lung Circ. 2014;23:905-912) used only improvement grade and rate of IMR. For determination of “recur”, the MR grade in perioperative TTE is to be described.
  • Post-Hoc analysis in the group of CABGa, the author use ERO > 10 mm2 as the recur of IMR. Is the valure related to outcome?
  • How about the post-hoc analysis in CABGmp group using the same multivariable analysis?

# Others

  • The viability of posterolateral wall is known to be important of IMR. And the table 2, the LCx treatment was significantly frequent. For the viability evaluation, four or more segment improvement in DBX is considered to be viable. How about focusing on the LCx territory viability?

Author Response

This study’s aim is to assess significance of echocardiographic parameters achieved from stress echo and low dose dbx as a predictor of recurrent IMR in patients with moderate or severe IMR qualified to CABG alone in 12-month follow-up. Patient selection criteria for combined CABG and MVP is a long-standing and important issue, and the use of stress echocardiography is considered as a useful method for selection. Some comments and question.

# Abstract

  • Please use full name of TA, CH in the abstract.

We change it.

  • The mitral deformation indices (MDI) generally include several parameters. Please specify parameters which are assessed in this study.

We assessed: tenting area, coaptation high and sphericity index.

# Introduction

  • Line 59, “approptiate surgical [11-13]”. Require correction.

We change “appropriate surgical” on “appropriate type of surgery”

# M&M

  • This study included not only moderate but also severe degree ischemic mitral regurgitation (IMR). This is a unique point. In the reference 19 and other studies, severe IMR generally goes to combination of MVP or other tract, and other studies included moderate IMR in the study population. I wonder the reason the authors include both moderate and severe IMR in the same population.

Answer: In some patients with severe mitral regurgitation, we have observed that during the dobutamine test there is a significant reduction in the degree of regurgitation and no increase in the degree of regurgitation during the exercise test. We hypothesized that in the group of patients with severe ischemic mitral regurgitation, there is a certain group of patients in whom it may be sufficient to perform CABG without mitral valve intervention. And we included sick patients in our analysis.

  • The Eligible criteria suggested by the authors have no reference. Please provide backgrounds of the eligible criteria. Other studies (Roshanali et al. Echocardiograhy 2006;23:31-37 and J Thorac Cardiovasc Surg 2014;148:1323-1327) used similar but more simple eligibility criteria.

Answer: We used such eligibility criteria based on our observations in our CABG qualified patients who underwent dobutamine stress tests and exercise tests in patients with mitral regurgitation. Therefore, the proposed qualification criteria for a given type of cardiac surgery treatment are our own.

  • Line 110, please use the same scale of ERO, (in some cases the authors used 10 mm2 and in some cases 0.1 cm2).
    We changed it: abbreviations have been standardized.
  • In the section of 2.3, “IMR severity was assessed by measuring the effective regurgitant orifice area (EROA), with EROA >10 mm² and <20 mm² considered moderate and ERO ≥20 mm² considered severe as well as mitral regurgitation volume (MRvol) with MR vol ≥ 30ml considered severe [17-19].” What is the background of ERO > 10 mm2 of moderate IMR?

Answer: According to the ESC standards in force at that time, severe ischemic mitral regurgitation was defined as ERO> = 0.2 cm2 and MRvol> = 30 ml. We introduced the term moderate mitral regurgitation (ERO 10-20 cm2) based on our observations of the resting echo, exercise echo and the results of dobutamine tests, as well as the occurrence of symptoms in patients with multivessel coronary disease qualified for CABG.

  • Line 117, EROA and ERO looks the same term. Yes. We changed it.
  • Line 129, “The mitral valve deformation” is the same term of “MDI?” Yes
  • Line 138, “DBX”, “dbx”, and “DSE” looks the same term. We changed (DBX is ok)
  • In the section of 2.4, Please describe the dobutamine doses for the DBX.

We added: “For detection of inotropic response in heart failure patients we used stages of 5 min, starting from 5 up to 20 mg/kg/min. (low dose DBX).”

# Results

  • In the table2, CABGa group has higher LCx lesions. IMR is known to be related to posterolateral wall involvement. Is the more LCx lesion related to poor outcome?

We didn’t analyze if more LCx lesion were related to poor outcome.

  • The author used “recurred” IMR. Was the IMR recurred or persisted after surgery? Other studies like Roshanali or Yin (Heart Lung Circ. 2014;23:905-912) used only improvement grade and rate of IMR. For determination of “recur”, the MR grade in perioperative TTE is to be described.

After CABG alone or with mitral annuloplasty, TTE evaluation revealed acceptable results in all the cases: there was no MR or mild MR. Before discharge, TTE was performed, which confirmed the good results of surgical treatment in both groups of patients. Recurrent IMR was the insufficiency of at least moderate (EROA>0.1 cm2) or more at follow-up visits.

Post-Hoc analysis in the group of CABGa, the author use ERO > 10 mm2 as the recur of IMR. Is the valure related to outcome?

The sample size of our study was too small to analyze this point.

  • How about the post-hoc analysis in CABGmp group using the same multivariable analysis?
  • Study population was too small to perform this type of analysis.

# Others

  • The viability of posterolateral wall is known to be important of IMR. And the table 2, the LCx treatment was significantly frequent. For the viability evaluation, four or more segment improvement in DBX is considered to be viable. How about focusing on the LCx territory viability?

In this analysis we didn’t focus only on the LCX territory viability

Round 2

Reviewer 1 Report

The authors have addressed just some of the issues from our previous review

  1. The title and purpose of the study is related to predictors of bad results regarding mitral regurgitation in patients submitted to just CABG. But then it appears the group CABG+mitral intervention, which is odd. I suggest changing the title to something like: “Stress echocardiography protocol for deciding type of surgery in ischemic mitral regurgitation. Predictors of bad results after CABG alone
  2. The main issue I would like to have clarified is: from patients with significant MR and LV dysfunction, how many were sent to CABGa or CABG plus mitral annuloplasty, based on Dobutamine and ExE results. Also, show what is going on with a patient that has severe MR with exercise and good deformation parameters with Dobutamine. Or the opposite situation?. How many patients were in these situation?. The total number of patients with the 4 possible scenarios with the decisions taken should be depicted (Dob+ plus ExE+; Dob- plus ExE -; and the 2 intermediate situations)
  3. State in the same paragraph of Methods or in Figure 1 how many patients were excluded due to absence of viability and the other types of exclusion the authors state some sentences further.
  4. Patients were selected for CABG with or without mitral intervention according to a specified protocol, but then those submited to CABG alone were studied to assess the predictors of MR recurrence. Therefore a limitation for these assessment was that only patients qualified for CABGa were included, meaning patients with no severe MR with exercise and with good deformation indexes with Dobutamine (according to Table 1)
  5. All the paragraph regarding Fig 6 is presented in a very disorganized. Specifically, the last sentence should go the first...
  6. Study inclusion/exclusion criteria: state if they were for the 2 subgroups or just CABGa.
  7. mitraclip intervention could be reserved for patients with CABGa in case of deterioration of an initially considered non severe MR.

8 State functional capacity in Watts.

  1. No sure if the assessment and presence of viability was necessary for inclusion in both groups or just the CBAGa group. Please clarify in methods
  2. How was presence of posterolateral viability related to follow-up MR results?. In the discussion section the authors mentioned about posterolateral wall viability findings but in vague manner (page 13)
  3. Abstract, conclusions: clarify and state that these findings were related to just the CABGa.
  4. Last paragraph of 2.2 Surgery: Some odd symbols appear here
  5. Discussion: the authors state that “complete revascularization resulted in significant improvement of LV geometry and function”, but I did not see any results regarding to complete or incomplete revascularization in these patients….

Author Response

Author’s reply

The authors have addressed just some of the issues from our previous review

  1. The title and purpose of the study is related to predictors of bad results regarding mitral regurgitation in patients submitted to just CABG. But then it appears the group CABG+mitral intervention, which is odd. I suggest changing the title to something like: “Stress echocardiography protocol for deciding type of surgery in ischemic mitral regurgitation. Predictors of bad results after CABG alone

Answer:

We agree.

we changed title:

“Stress echocardiography protocol for deciding type of surgery in ischemic mitral regurgitation. Predictors of mitral regurgitation recurrence following CABG alone.”

2. The main issue I would like to have clarified is: from patients with significant MR and LV dysfunction, how many were sent to CABGa or CABG plus mitral annuloplasty, based onDobutamine and ExE results.

Answer:

We have clarified these data in the “Methods” section:

“Of a total of 170 potentially eligible patients with IMR qualified to CABG were enrolled in the present study. …….
Finally, one hundred forty-five patients were sent to CABGa or CABGmp and prospectively enrolled into the study.

Also, show what is going on with a patient that has severe MR with exercise and good deformation parameters with Dobutamine. Or the opposite situation?. How many patients were in these situation?. The total number of patients with the 4 possible scenarios with the decisions taken should be depicted (Dob+ plus ExE+; Dob- plus ExE -; and the 2 intermediate situations).

Answer:

We modified table 1: we included all Eligibility criteria for the appropriate surgery method surgery (all scenarios of qualification process). We included additional group of patients sent to th CABG with mitral replacement (CABGmr), but this group was excluded from the further analysis.

3. State in the same paragraph of Methods or in Figure 1 how many patients were excluded due to absence of viability and the other types of exclusion the authors state some sentences further.

Answer:

We have clarified these data in the “Methods” section:

Of a total of 170 potentially eligible patients with IMR qualified to CABG were enrolled in the present study. Of these, 25 patients were excluded from the analysis (7 patients had poor acoustic window for dobutamine echocardiography, 4 patients in whom viability was not demonstrated within left ventricular segments with impaired contractility, 14 had contraindications for exercise).”

4. Patients were selected for CABG with or without mitral intervention according to a specified protocol, but then those submited to CABG alone were studied to assess the predictors of MR recurrence. Therefore, a limitation for these assessment was that only patients qualified for CABGa were included, meaning patients with no severe MR with exercise and with good deformation indexes with Dobutamine (according to Table 1).

Answer:

Yes, we present the entire qualification process (all criteria and possible scenarios), but in this publication we focused only on the CABG group and the problem of recurrent mitral regurgitation in this group (by the way, the most numerous). We mention other patient groups (CABGmp and CABG mr) in the methodology to accurately present the patient qualification process.

5. All the paragraph regarding Fig 6 is presented in a very disorganized. Specifically, the last sentence should go the first...

Answer:

We have made significant corrections to the "Echocardiographic Results / Recurrence Mitral Regurgitation" section, especially in the section on the description of Tables 5 and 6.

6. Study inclusion/exclusion criteria: state if they were for the 2 subgroups or just CABGa.

Answer:

Study inclusion/exclusion criteria were for both study groups (CABG alone and CABG mp)

7. mitraclip intervention could be reserved for patients with CABGa in case of deterioration of an initially considered non severe MR.

Answer:

We fully agree with this statement.

Our study was done a few years ago when mitraclip interventions were not as available as they are today.

Currently, in patients with recurrence of significant IMR after CABG surgery, mitraclip may be a good option.

  1. State functional capacity in Watts.

Answer

We added it in tables 3 and 6.

9. No sure if the assessment and presence of viability was necessary for inclusion in both groups or just the CBAGa group. Please clarify in methods

Answer:

Yes, myocardial viability was necessary for inclusion in both study groups.

So, in the "Study population" section, we added a sentence:

“The assessment and presence of myocardial viability was necessary for inclusion in both study groups. “

10. How was presence of posterolateral viability related to follow-up MR results?. In the discussion section the authors mentioned about posterolateral wall viability findings but in vague manner (page 13)

Answer:

In the discussion section, we mentioned the findings on the viability of the posterolateral wall, but we did not analyze the relationship between the viability of selected LV segments (i.e., postero-lateral wall) and IMR recurrence. In the discussion section, we note the possibility of such a dependence.

11. Abstract, conclusions: clarify and state that these findings were related to just the CABGa.

Answer:

We changed: Preoperative assessment of MDI changes during dbx can be used to identify patients with IMR qualified to CABG alone at increased risk of recurrence of IMR in 1 year follow-up.

12. Last paragraph of 2.2 Surgery: Some odd symbols appear here

We checked the last paragraph of 2.2 Surgery: we didn't notice any extra / odd symbols

13. Discussion: the authors state that “complete revascularization resulted in significant improvement of LV geometry and function”, but I did not see any results regarding to complete or incomplete revascularization in these patients….

Answer:

Ultimately, only patients who managed to complete revascularization were included in the study. Only such patients were included in the further analysis.

So in the "Study population" section, we added a sentence:

“Only patients who underwent complete revascularization were included in the further analysis.”

Reviewer 2 Report

Dear Authors,

Thank you for your reply and revision. I’d like to comment on the reply of the authors.

  • This study included not only moderate but also severe degree ischemic mitral regurgitation (IMR). This is a unique point. In the reference 19 and other studies, severe IMR generally goes to combination of MVP or other tract, and other studies included moderate IMR in the study population. I wonder the reason the authors include both moderate and severe IMR in the same population.

Answer: In some patients with severe mitral regurgitation, we have observed that during the dobutamine test there is a significant reduction in the degree of regurgitation and no increase in the degree of regurgitation during the exercise test. We hypothesized that in the group of patients with severe ischemic mitral regurgitation, there is a certain group of patients in whom it may be sufficient to perform CABG without mitral valve intervention. And we included sick patients in our analysis.

  • The Eligible criteria suggested by the authors have no reference. Please provide backgrounds of the eligible criteria. Other studies (Roshanali et al. Echocardiograhy 2006;23:31-37 and J Thorac Cardiovasc Surg 2014;148:1323-1327) used similar but more simple eligibility criteria.

Answer: We used such eligibility criteria based on our observations in our CABG qualified patients who underwent dobutamine stress tests and exercise tests in patients with mitral regurgitation. Therefore, the proposed qualification criteria for a given type of cardiac surgery treatment are our own.

I agreed that your work is based on the criteria of your own. And the criteria used in reference 13 (Roshanali) are very similar with yours. Would you add some discussion on this point? That will be support for your criteria.

Author Response

Dear Reviewer,

Your comment:

“I agreed that your work is based on the criteria of your own. And the criteria used in reference 13 (Roshanali) are very similar with yours. Would you add some discussion on this point? That will be support for your criteria.”

Thank you for your valuable comment.

In line with your review, we have added a short comment on the echocardiographic criteria for eligibility for surgery that we used in the protocol of our study. Thank you for your reply and revision. I’d like to comment on the reply of the authors.

In Discussion section we added:

“Roshanali et al. used only simple qualitative echocardiographic criteria based on the 4-grade scale of mitral regurgitation to identify patients with moderate IMR. They showed the utility of low-dose DBX in selecting patients who would be undergoing CABG to receive concurrent mitral valve repair {13}. We have developed a more advanced protocol based on exercise echocardiography (ExE) and dobutamine stress echocardiography (DBX) criteria for precise determination of the range of surgical interventions.”
